# Exploring if and how evidence-based practice of occupational and physical therapists evolves over time: A longitudinal mixed methods national study

**Muhammad Zafar Iqbal**[1,2], **Annie Rochette**[3,4], **Nancy E. Mayo**[1,5], **Marie-France Valois**[6], **André E. Bussières**[7], **Sara Ahmed**[1,2,5], **Richard Debigaré**[8], **Lori Jean Letts**[9], **Joy C. MacDermid**[10], **Tatiana Ogourtsova**[1,2,11], **Helene J. Polatajko**[12], **Susan Rappolt**[12], **Nancy M. Salbach**[13], **Aliki Thomas**[1,2,14]*

1 School of Physical and Occupational Therapy, Faculty of Medicine and Health Sciences, McGill University, Montreal, Quebec, Canada, 2 Centre for Interdisciplinary Research in Rehabilitation of Greater Montreal (CRIR), Montreal, Quebec, Canada, 3 School of Rehabilitation, Université de Montréal, Montreal, Quebec, Canada, 4 Centre for Interdisciplinary Research in Rehabilitation, Institut universitaire sur la réadaptation en déficience physique de Montréal (IURDPM), Montreal, Quebec, Canada, 5 Center for Outcomes Research and Evaluation (CORE), Research Institute of McGill University, Montreal, Quebec, Canada, 6 Department of Medicine, Faculty of Medicine, McGill University, Montreal, Quebec, Canada, 7 Département Chiropratique, Université du Québec à Trois-Rivières, Trois-Rivières, Quebec, Canada, 8 Centre de recherche de l'Institut universitaire de cardiologie et de pneumologie de Québec, Université Laval, Laval, Quebec, Canada, 9 School of Rehabilitation Sciences, Faculty of Health Sciences, McMaster University, Hamilton, Ontario, Canada, 10 School of Physical Therapy and Department of Surgery, University of Western Ontario, London, Ontario, Canada, 11 Research Center of the Jewish Rehabilitation Hospital, Centre intégré de santé et de services sociaux de Laval, Laval, Quebec, Canada, 12 Department of Occupational Science and Occupational Therapy, Rehabilitation Sciences Institute, University of Toronto, Toronto, Ontario, Canada, 13 Department of Physical Therapy, Rehabilitation Sciences Institute, University of Toronto, Toronto, Ontario, Canada, 14 Institute of Health Sciences Education, McGill University, Montreal, Quebec, Canada

* aliki.thomas@mcgill.ca

**Data Availability Statement:** We did not obtain permission from the Institutional Review Board of McGill University for public availability of the data,

## Abstract

### Background

Occupational therapists (OTs) and physiotherapists (PTs) are expected to provide evidence-based services to individuals living with disabilities. Despite the emphasis on evidence-based practice (EBP) by professional entry-level programs and professional bodies, little is known about their EBP competencies upon entry to practice and over time or what factors impact EBP use. The aim of the study was to measure and understand how EBP evolves over the first three years after graduation among Canadian OTs and PTs, and how individual and organizational factors impact the continuous use of EBP.

### Methods

A longitudinal, mixed methods sequential explanatory study. We administered a survey questionnaire measuring six EBP constructs (knowledge, attitudes, confidence, resources, use of EBP and evidence-based activities) annually, followed by focus group discussions with a subset of survey participants. We performed group-based trajectory modeling to

nor did we obtain consent from participants. Therefore, we cannot share or make the data public. However, raw data are available from the corresponding author for verification purpose, or upon request from McGill (https://www.mcgill.ca/drs/rdm).

**Funding:** AT (Principal Investigator) received the research grant for this research project. The study was supported by grant from the Canadian Institute of Health Research (CIHR), Grant Number: 148544 https://cihr-irsc.gc.ca/e/193.html CIHR had no role in study design, data collection and analysis, decision to publish, or preparation of the manuscript.

**Competing interests:** The authors have declared that no competing interests exist.

identify trajectories of EBP over time, and a content analysis of qualitative data guided by the Theoretical Domains Framework.

## Results

Of 1700 graduates in 2016–2017, 257 (response rate = 15%) responded at baseline (T0) (i.e., at graduation), and 83 (retention rate = 32%), 75 (retention rate = 29%), and 74 (retention rate = 29%) participated at time point 1 (T1: one year into practice), time point 2 (T2: two years into practice, and time point 3 (T3: three years into practice) respectively. Group-based trajectory modeling showed four unique group trajectories for the use of EBP. Over 64% of participants (two trajectories) showed a decline in the use of EBP over time. Fifteen practitioners (7 OTs and 8 PTs) participated in the focus group discussions. Personal and peer experiences, client needs and expectations, and availability of resources were perceived to influence EBP the most.

## Conclusions

Though a decline in EBP may be concerning, it is unclear if this decline is clinically meaningful and whether professional expertise can offset such declines. Stakeholder-concerted efforts towards the common goal of promoting EBP in education, practice and policy are needed.

## Introduction

Occupational therapists (OTs) and physiotherapists (PTs) provide services to individuals and communities with a primary objective to improve quality of life, wellbeing, and participation in daily activities. While making decisions about patient care, these practitioners are expected to combine the best available scientific evidence with their clinical expertise and patient preferences to achieve optimal, satisfactory and cost-effective outcomes [1–3], a well-known process referred to as evidence-based practice (EBP) [4].

Enactment of EBP in the clinical and/or practice context can be operationalized through the self-reported use of EBP and indirectly through the number and type of evidence-based (EB) activities done in daily practice [5]. In our previous work, use of EBP was defined as "the actual application of EBP concepts, tools, and procedures into specific actions" [5] (p.3). Identifying a gap in knowledge related to a patient situation, effectively conducting an online literature search to address the research question, and critically appraising the strengths and weaknesses of study methods are some examples of use of EBP. Evidence-based (EB) activities refers to "the implementation of research evidence to the surrounding environment" [5] (p.3). Informally sharing and discussing literature/research findings with colleagues or patients, integrating research evidence with expertise, and making time and reading research reports are examples of EB activities. In other words, use of EBP is the actual engagement of the practitioner in the EBP process, whereas EB activities serve as a medium to facilitate the use of EBP.

As a complex decision-making approach, EBP in rehabilitation services has been the subject of significant theoretical and empirical examination over the past three decades [6–9]. Because of the "promise" of EBP and its potential to positively impact patient outcomes, national professional associations such as the *Canadian Association of Occupational Therapists* and the *Canadian Physiotherapy Association* have urged OT and PT practitioners through position

statements to embrace and apply EBP [10, 11]. Consequently, all Canadian professional OT and PT graduate programs have incorporated EBP as a core competency in their professional entry-to-practice programs [12].

Despite an increased emphasis on EBP reflected in major changes in professional curricula in Canada and elsewhere in the world, the use of EBP by OT and PT practitioners remains a challenge [13–15]. This is somewhat concerning when studies show that early career practitioners generally hold positive attitudes towards EBP [14, 16, 17], but that only half use EBP [13]. Findings from our nationwide study showed that two-thirds of the participating OT and PT graduates reported using EBP upon entry to practice [5]. In studies including different health professions, PTs showed more positive attitudes towards EBP but lesser use of EBP than other professions such as physicians, nurses, podiatry and radiology [18, 19]. Similarly, a moderate use of EBP was found in a survey of more than 1500 OTs in New Zealand [20]. In parallel, studies of engagement of practitioners in EB activities report inconsistent findings, ranging from reasonable [5, 13, 15] to suboptimal [20, 21] involvement in such activities.

One reason for the variable use of EBP and EB activities among early career OT and PT practitioners appears to be the complex and multidimensional nature of the EBP process [17]. Evidence-based practice depends upon one's ability to mobilize several individual factors (e.g., knowledge, attitudes, and confidence) [21–23]. Challenges in critically appraising the scientific literature, a lack of confidence in applying research to practice, time constraints, and varying attitudes about the relevance and applicability of scientific evidence in practice account for some of the most salient factors highlighted in the literature [24–27]. Moreover, many studies suggest that practitioners often prefer personally "tried and tested" methods over empirical evidence while providing patient care [3, 20, 15, 28, 29], which has been identified as a barrier to the enactment of EBP. In addition to the individual factors, EBP also depends upon practitioners' ability to navigate numerous organizational (e.g., resources) [30, 31] and systemic (e.g., productivity pressure, policy obligations, and lawsuits) [32–34] factors. For instance, lack of resources (e.g., funding, leadership support, recognition, continuing professional development activities, access to literature), high patient caseload, reduced peer interactions during case discussions, and poor role modeling have been shown to negatively influence the uptake and utilization of evidence in clinical practice [24, 29, 35, 36].

Many studies have explored the influence of individual and/or organizational factors on EBP in different professional contexts (e.g., medicine, nursing, OT and PT) [15, 17, 25, 26, 37, 38]. Most of these studies have been conducted at a specific time point or with experienced practitioners; the data from these studies cannot be used to predict the transition of EBP among the early career practitioners, all of whom were highly trained in EBP. Moreover, most previous studies were either small scale, single site studies or lacked robust analytical approaches and measures [17].

There was, therefore, a need for a longitudinal nationwide study to track EBP and its associated factors among early career OT and PT practitioners. Longitudinally examining if and how EBP changes over time, and which factors influence this change might support curricular reforms of professional OT and PT programs across Canada. This exploration may also inform future knowledge translation interventions designed to positively influence EBP competencies.

## Objectives

To measure and understand how EBP evolves over the first three years after graduation among Canadian OTs and PTs and how individual and organizational factors impact the continuous use of EBP.

## Materials and methods

### Study design

We conducted a longitudinal, cohort-based, mixed methods sequential explanatory study [39, 40] spanning a period of three years (2016–17 until 2020–21). A sequential explanatory mixed methods study design, grounded in a postpositivist paradigm, is a methodology for sequentially collecting, analyzing, and interpreting the quantitative and qualitative data in a single study to synergistically investigate the same underlying phenomenon or research question [39, 40]. More specifically, we used a *fully mixed sequential dominant status design* that mixes quantitative and qualitative research within one, or across different stages of the research process, but one component (either quantitative or qualitative) remains dominant and leads to the design of the other component for further exploration [41]. In our case, the quantitative phase was the dominant component as it led to the identification of the EBP trajectories (primary study objective), whereas the qualitative phase was sequentially (after each survey data collection time) integrated throughout the duration of the study to deepen our understanding of the individual and organizational facilitators and barriers to EBP.

### Phase 1 (quantitative)

**Participants.** We targeted the graduates of 2016–17 (n = 1703) of all 29 OT and PT programs across Canada for our longitudinal data collection.

**Procedure.** The quantitative data were collected at four time points (T0, T1, T2, and T3) from OT and PT graduates of the professional M. Sc. programs from the time of graduation up to three years into clinical practice. The baseline data (T0) were collected in 2016–2017 from those who graduated in the last six months because programs had staggered graduation dates. Time points T1, T2 and T3 represent the data collected after one, two and three years into clinical practice, respectively. Since we wanted to observe the pattern of EBP over time, only those OT and PT graduates were eligible to participate in the subsequent rounds (T1, T2 and T3) who had participated in the baseline survey and were working at the time of data collection.

Within six months of graduation, we reached out to all new graduates from 28/29 OT and PT programs for baseline (T0) data collection. In our introductory email, we explained the study objectives and provided an electronic link to complete the study questionnaire on a web-based survey platform (Lime Survey). The data at T1, T2 and T3 were collected using the same procedure.

**Instrumentation.** More than 100 measures exist that quantify different factors, either alone or in combination, affecting EBP [42–44]. However, none of these are designed to measure the breadth of individual and organizational constructs related to EBP. We developed a comprehensive measure to address this gap that is described in detail elsewhere [45, 46]. The measure captures six constructs: knowledge (8 items), attitudes (10 items), confidence (i.e., self-efficacy, 8 items), organizational resources (13 items), use of EBP (9 items), and EB activities (7 items) [46]. Questions related to demographic characteristics were also included in the survey (see Table 1). The questionnaire was available in both English and French languages.

**Data analysis.** To longitudinally measure EBP and how it evolves over time, we used group-based trajectory modeling for all six EBP constructs [47]. Group-based trajectory modeling is a form of latent class analysis that is used to identify distinctive groups with common trajectories of change among groups of well-characterized individuals [47, 48]. We rescaled all reflective constructs (knowledge, attitudes, confidence, resources) from their original scales to a 0–100 range. In the trajectory analysis, we included data of those participants who participated in at least two time points. Since our objective was to measure actual use of

**Table 1. Time points wise characteristics of the participants including raw scores on the six EBP constructs measured.**

| | T0 | T1 | T2 | T3 |
|---|---|---|---|---|
| | (N = 257) | (N = 83) | (N = 75) | (N = 74) |
| **Gender, n (%)** | | | | |
| Female | 214 (83%) | 69 (83%) | 66 (88%) | 67 (91%) |
| Male | 39 (15%) | 11 (13%) | 6 (8%) | 5 (7%) |
| Prefer not to answer | 1 (0%) | 1 (1%) | 1 (1%) | 1 (1%) |
| Missing | 3 (1%) | 2 (2%) | 2 (3%) | 1 (1%) |
| **Language of survey completion, n (%)** | | | | |
| English | 181 (70%) | 58 (70%) | 55 (73%) | 54 (73%) |
| French | 76 (30%) | 25 (30%) | 20 (27%) | 20 (27%) |
| **Profession of participants, n (%)** | | | | |
| Occupational therapy | 135 (53%) | 44 (53%) | 39 (52%) | 47 (64%) |
| Physical therapy | 122 (47%) | 39 (47%) | 36 (48%) | 27 (36%) |
| **Current clinical setting (if working) [†], n (%)** | | | | |
| Private Practice | 92 (63%) | 27 (33%) | 26 (36%) | 1 (1%) |
| General hospital–acute care | 14 (10%) | 18 (22%) | 10 (14%) | 14 (20%) |
| Home Visiting Agency | 14 (10%) | 4 (5%) | 6 (8%) | 0 (0%) |
| Community Agency | 9 (6%) | 8 (10%) | 8 (11%) | 10 (15%) |
| Rehabilitation Centre | 5 (3%) | 15 (18%) | 13 (18%) | 11 (16%) |
| Primary Health Care | 3 (2%) | 2 (2%) | 3 (4%) | 5 (7%) |
| Long-term care / Complex continuing care | 3 (2%) | 0 (0%) | 3 (4%) | 3 (4%) |
| General hospital–long term care | 2 (1%) | 2 (2%) | 1 (1%) | 1 (1%) |
| Consulting Firm | 2 (1%) | 0 (0%) | 0 (0%) | 20 (29%) |
| Missing* | 3 (2%) | 6 (7%) | 2 (3%) | 4 (6%) |
| **Number of hours worked per week as an OT or PT (if working), n (%)** | | | | |
| >35 | 52 (35%) | 48 (59%) | 50 (70%) | 36 (52%) |
| 29–35 | 53 (36%) | 24 (29%) | 16 (22%) | 25 (36%) |
| 22–28 | 14 (10%) | 3 (4%) | 4 (6%) | 4 (6%) |
| 15–21 | 15 (10%) | 3 (4%) | 1 (1%) | 3 (4%) |
| 8–14 | 5 (3%) | 0 (0%) | 1 (1%) | 0 (0%) |
| <7 | 8 (6%) | 2 (2%) | 0 (0%) | 0 (0%) |
| Missing* | 0 (0%) | 2 (2%) | 0 (0%) | 1 (2%) |
| **Number of patients/clients seen in a typical day (if working), n (%)** | | | | |
| 10 or more | 37 (25%) | NA | 24 (33%) | 12 (17%) |
| 6–9 | 44 (30%) | NA | 23 (32%) | 23 (33%) |
| 3–5 | 46 (31%) | NA | 21 (29%) | 27 (39%) |
| 2 or less or do not provide services to clients/patients | 20 (14%) | NA | 4 (6%) | 6 (9%) |
| Missing* | 0 (0%) | NA | 0 (0%) | 1 (2%) |
| **Number of same profession therapists in your workplace including yourself (if working), n (%)** | | | | |
| 11 or more | 22 (15%) | 23 (28%) | 26 (36%) | 17 (25%) |
| 7–10 | 9 (6%) | 0 (0%) | 9 (12%) | 11 (16%) |
| 4–6 | 8 (5%) | 0 (0%) | 15 (21%) | 16 (23%) |
| 3 or less | 57 (39%) | 25 (30%) | 22 (31%) | 24 (35%) |
| Missing* | 51 (35%) | 34 (42%) | 0 (0%) | 1 (1%) |
| **Continuing education since obtaining professional license (if working), n (%)** | | | | |
| No | 84 (57%) | 6 (7%) | 1 (1%) | 1 (1%) |
| Yes | 63 (43%) | 76 (93%) | 71 (99%) | 67 (97%) |
| Missing* | 0 (0%) | 0 (0%) | 0 (0%) | 1 (1%) |

*(Continued)*

**Table 1.** (Continued)

| | T0 | T1 | T2 | T3 |
|---|---|---|---|---|
| | (N = 257) | (N = 83) | (N = 75) | (N = 74) |
| **University affiliated setting (if working), n (%)** | | | | |
| No | 91 (62%) | 43 (52%) | 47 (65%) | 38 (55%) |
| Yes | 34 (23%) | 32 (39%) | 23 (32%) | 26 (38%) |
| I don't know | 22 (15%) | 7 (9%) | 2 (3%) | 4 (6%) |
| Missing* | 0 (0%) | 0 (0%) | 0 (0%) | 1 (1%) |
| **Work as part of an interdisciplinary team (if working), n (%)** | | | | |
| Yes | 115 (78%) | 69 (84%) | 58 (81%) | 57 (83%) |
| No | 32 (22%) | 13 (16%) | 14 (19%) | 11 (16%) |
| Missing | 0 (0%) | 0 (0%) | 0 (0%) | 1 (1%) |
| **Rasch Measurements (Scales)** | | | | |
| Use of EBP (5–9) | | | | |
| Mean (SD) | 7.2 (1.5) | 7.1 (1.6) | 6.7 (1.5) | 6.6 (1.5) |
| [Min–Max] | [5 – 9] | [5 – 9] | [5 – 9] | [5 – 9] |
| 95% Cl | 7.0–7.5 | 6.8–7.4 | 6.3–6.9 | 6.4–7.1 |
| EB Activities (0–140) | 18.3 (20.2) | NA* | 12.7 (10.6) | 12.7 (11.5) |
| | [0–122] | | [0 – 49] | [1 – 70] |
| | 15.0–21.7 | | 10.2–15.2 | 9.9–15.5 |
| Knowledge (0–29) | 21.8 (4.7) | 20.5 (5.3) | 20.4 (5.1) | 20.1 (4.6) |
| | [8 – 29] | [8 – 29] | [12 – 29] | [13 – 29] |
| | 21.2–22.4 | 19.4–21.7 | 19.2–21.5 | 19.3–21.5 |
| Attitude (0–32) | 22.0 (3.7) | 20.1 (3.1) | 20.3 (3.5) | 20.2 (3.2) |
| | [11 – 31] | [12 – 26] | [12 – 29] | [10 – 28] |
| | 21.5–22.4 | 19.4–20.8 | 19.5–21.1 | 19.6–21.1 |
| Confidence (0–22) | 13.5 (4.2) | 11.8 (4.6) | 13.8 (4.0) | 13.8 (4.1) |
| | [0 – 22] | [1 – 22] | [3 – 22] | [2 – 22] |
| | 12.9–14.0 | 10.8–12.8 | 12.8–14.7 | 12.8–14.8 |
| Resources (0–39) | 25.4 (5.7) | 24.3 (6.0) | 22.6 (5.7) | 23.7 (5.5) |
| | [8 – 39] | [10 – 38] | [10 – 36] | [12 – 37] |
| | 24.4–26.4 | 23.0–25.6 | 21.2–24.0 | 22.3–25.0 |

T0: Time point at which baseline data was collected within 6 months of graduation

T1: Time point at which data was collected after one year into clinical practice

T2: Time point at which data was collected after two years into clinical practice

T3: Time point at which data was collected after three years into clinical practice

NA: Not Applicable or Not Available

*Represents those who did not respond to these specific questions as responses were on voluntary basis

†Questions related to employment were only answered by those working at the time of the survey.

CI = Confidence Interval

EBP, we included responses of only those participants who were working at the time of data collection; their first job was considered as the baseline data. For instance, if a participant was not working at T0 but participated at two other time points (e.g., T1 and T2), we treated their responses at T1 as baseline data in the trajectory analysis. Those who responded at baseline only were excluded from the trajectory analysis. Censored normal model was then used on the included data to identify distinctive groups of OT and PT practitioners showing similar trajectories of EBP over three years (four time points).

Using baseline data as a reference point, different models were compared, including models with different number of trajectories as well as with different shapes of the same trajectory (intercept only, linear, and quadratic shape). We used the Bayesian Information Criteria (BIC), Akaike's Information Criteria (AIC) and posterior probability to assess model fit and best model selection [47]. All trajectory analyses were performed by an expert statistician (MFV) using the Statistical Analysis Software (SAS)$^{\circledR}$, PROC TRAJ (Version 9.4) developed by Jones and Nagin [49].

## Phase 2 (qualitative)

We used the *Consolidated criteria for reporting qualitative research (COREQ)* criteria [50] for reporting the qualitative phase. Detailed report is given in S1 Appendix.

**Participants.** Using a purposive sampling technique [51], we recruited OT and PT practitioners from those who volunteered in the survey [5] to participate in a focus group discussion.

**Procedure.** We conducted focus groups at T1, T2 and T3 using a discussion guide (see S2 Appendix) based on 1) the Theoretical Domains Framework (TDF) [52]; 2) the results from the quantitative analyses,; and 3) our previously published work [53]. The TDF is a comprehensive determinant framework grounded in 33 behavioral change theories. It includes 14 domains: *knowledge*, *beliefs about capabilities*, *behavioral regulation*, *skills*, *beliefs about consequences*, *environmental context and resources*, *social influences*, *social/professional role and identity*, *emotions*, *goals*, *decision processes*, *reinforcement*, *optimism*, and *intention* [52]. The framework offers a theoretical lens for identifying the cognitive, affective, social and environmental factors that influence human behavior related to the implementation of EBP [52, 54]. Two team members (AT and AR) developed the first version of the guide. A TDF expert (AB) then reviewed it for face and content validity, ensuring that the questions were clear, easy to understand and covered each domain. We probed for concrete examples from participants' clinical practice for illustration and clarification. The recorded discussions were transcribed verbatim.

**Data analysis.** We performed a conventional content analysis [55] of the focus group data using NVivo (QRS International, Melbourne, Australia) version 12. Following the recommended approach to analyzing TDF data [54], a qualitative researcher (TO) deductively coded each utterance into relevant TDF domains. The codes were then reviewed by two investigators (AT and AR) and a third investigator (AB) was invited to resolve conflicts in assigning utterances into their relevant domains. Any utterance not fitting into any of the 14 TDF domains was separately categorized into an additional domain called 'other'. We then linked utterance responses to specific beliefs. A specific belief is a statement that provides details about the role of the domain in influencing the behavior [52, 56]. These statements are intended to convey a meaning that is common to multiple utterances. We generated specific beliefs from utterances that captured the core thought and continued this process for every utterance. Beliefs, coded as being similar or identical statements, were then grouped according to their likelihood to increase (i.e., perceived to facilitate EBP), decrease (i.e., perceived barriers to EBP) or have no influence on the behavior. Finally, we identified which of the 14 domains were most relevant based on three criteria: presence of conflicting beliefs, strong beliefs perceived to impact EBP behavior and high frequency of specific beliefs. All three criteria were weighed equally to judge relevance of the domains as they relate to influencing EBP behavior.

## Ethical considerations

The ethical approval for this study was received from the Institutional Review Board of McGill University (IRB # A10-B55-16B). All participants provided consent before taking the survey and/or participating in focus group discussions.

# Results

## Phase 1 (quantitative)

Graduates from 28 out of 29 programs participated in this study. Of the 257 OTs and PTs (response rate = 15%) who responded at baseline (T0), 83, 75 and 74 participated at T1, T2 and T3 time points, representing retention rates of 32%, 29% and 29%, respectively. The majority of respondents were females, anglophones, predominantly working in private practice settings at baseline. Table 1 shows the time point wise demographic data of the participants and raw scores on the six EBP constructs measured.

**Group trajectories of EBP constructs.** Of the 257 participants, 100 (39%) participated in at least two time points, where: 39 participated at two time points, 42 at three time points and 19 at all four time points. We conducted 14 cross-comparative analyses and chose the best model that had the smallest BIC-A, BIC-B, AIC-A and AIC-B values. The differences between the theoretical proportions and calculated group size across groups were small (0.1% to 1.2%), which indicates that the classification errors were minimal. Finally, the mean posterior probabilities by group were found to be high for all EBP constructs. More specifically, the posterior probability was 0.77–0.91 for the use of EBP; 0.92–0.98 for EB activities; 0.77–0.96 for knowledge; 0.80–0.85 for attitudes; 0.92–0.96 for confidence; and 0.85–0.88 for resources. All these statistics indicate a good fit of the chosen model.

The best fit model identified different distinctive groups for different EBP constructs. For instance, we found four group trajectories for the use of EBP as the primary outcome. We labelled trajectories by the level of EBP at the start and end of the trajectory. Group 1 participants (15.1%) showed *medium to low* use of EBP, meaning that they started with mid-range use of EBP and decreased over time. Group 2 participants (19.9%) showed *low to medium* use of EBP, meaning that they started with very low use of EBP and gradually increased over time but never reached the highest level. Group 3 participants (48.8%) showed *medium-high to medium-low* use of EBP, meaning that they slightly decreased their use of EBP over time. Finally, Group 4 participants (16.2%) showed *consistently high* (stays flat) use of EBP, meaning that they started clinical practice with a high use of EBP and remained consistently high over time. We found four unique trajectories for knowledge, but three for attitudes, and two for EB activities, confidence and resources within the data included in the group-based trajectory modeling (see Fig 1).

The trajectory analysis (see Fig 1) shows a sample characterized by overall high level of EB activities (81%), a variety in level of knowledge (four relatively stable trajectories), relatively low attitudes (83.9%), a dichotomy in confidence (54.1% being low), and relatively high resources (63.9%). The differences between the theoretical proportions and calculated group size across groups were small which is why we decided to present the theoretical proportions in tabular form. Table 2 shows cross tabulations of prevalence of use of EBP (columns) with EB activities, knowledge, attitudes, confidence, and resources (rows). From these analyses, we observe that a low level of EB activities, a low confidence level and a mid-level of knowledge are almost sufficient to not have a high use of EBP. The strongest way to have a high use of EBP is to have highly positive attitudes towards EBP. However, having high levels of EB activities, attitudes, confidence, resources, and knowledge does not appear to be sufficient to avoid a decrease in the use of EBP over time (Group 3).

## Phase 2 (qualitative data)

Fifteen clinicians (9 OTs and 6 PTs) participated in the focus groups: six (3 OTs, 3 PTs) at T1, three OTs at T2 and six (3 OTs, 3 PTs) at T3. Focus group participants' average age was

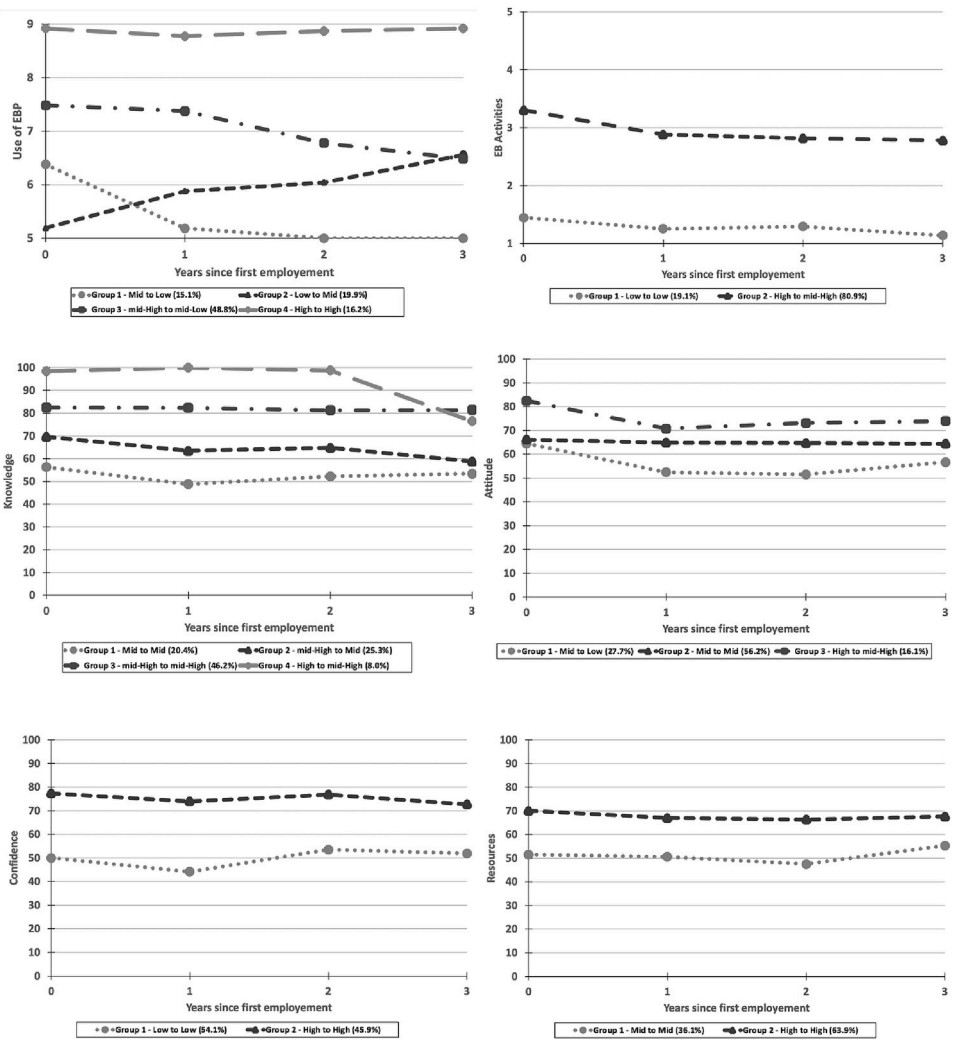

**Fig 1. Over the time change in group trajectories of OTs and PTs in all six EBP constructs from the time of first employment.**

26.2 ± 1.9 and 87% (13/15) were females. Two of the 15 worked in a hospital, one in a rehabilitation center, four in a local community services center, two in a home visiting agency, and six in private practice. With respect to the trajectory of use of EBP, one participant belonged to Group 1 (mid to low use of EBP), two belonged to Group 2 (low to mid use of EBP), ten belonged to Group 3 (mid-high to mid-low use of EBP), and two belonged to Group 4 (high to high use of EBP).

Our coding identified a total of 519 utterances, representing 233 beliefs across three time points. Of the 519 utterances, 67% emerged as facilitators for EBP. S3 Appendix provides time point wise numbers and percentages of domain specific utterances and beliefs (see Supporting information). The content analysis of all three time points collectively showed that all TDF domains were perceived as facilitators of EBP, except for 'environmental context and resources' and 'emotions' that were reported as barriers. S4 Appendix provides a detailed time point description of specific beliefs and sample quotes assigned to the relevant domains.

**Table 2. Cross tabulations of prevalence of use of EBP with EB activities, knowledge, attitudes, confidence, and resources.**

| | Line on Fig 1 (%) | Use of EBP | | | |
|---|---|---|---|---|---|
| | | Group 1 | Group 2 | Group 3 | Group 4 |
| | | Mid to Low | Low to Mid | Mid-High to Mid-Low | High to High |
| | | (N = 15) | (N = 19) | (N = 50) | (N = 16) |
| **EB activities** | | | | | |
| Low to Low | 1 (19.1%) | 47 | 32 | 21 | 0 |
| High to mid-High | 2 (80.9%) | 7 | 16 | 57 | 20 |
| **Knowledge** | | | | | |
| Mid to Mid | 1 (20.4%) | 30 | 25 | 40 | 5 |
| mid-High to Mid | 2 (25.3%) | 8 | 28 | 60 | 4 |
| mid-High to mid-High | 3 (46.2%) | 15 | 11 | 51 | 23 |
| High to mid-High | 4 (8.0%) | 0 | 25 | 37 | 37 |
| **Attitude** | | | | | |
| Mid to Low | 1 (27.7%) | 28 | 16 | 48 | 8 |
| Mid to Mid | 2 (56.2%) | 12 | 21 | 54 | 13 |
| High to mid-High | 3 (16.1%) | 7 | 14 | 36 | 43 |
| **Confidence** | | | | | |
| Low to Low | 1 (54.1%) | 17 | 23 | 51 | 9 |
| High to High | 2 (45.9%) | 12 | 14 | 49 | 25 |
| **Resources** | | | | | |
| Mid to Mid | 1 (36.1%) | 21 | 30 | 30 | 18 |
| High to High | 2 (63.9%) | 12 | 13 | 60 | 15 |

Notes:

• Rows may not add to 100% exactly because of rounding.

• The top row of the table represents the proportions of the four trajectories on the use of EBP.

• The second column presents the theoretical proportions within the trajectories of EBP knowledge, attitudes, confidence, and resources.

• The body of the table represents conditional proportions.

## Discussion

In this longitudinal, mixed-method study that spanned a period of three years, we sought to explore if and how EBP evolves among OT and PT graduates entering practice and what factors are associated with their use of EBP over time. There are four key findings from the quantitative data: (1) a slight but steady decrease in the use of EBP amongst two thirds of the trajectory participants; (2) those who started with high use of EBP after graduation continued to do so over time; (3) only a small subset showed an increase in the use of EBP; and (4) among all EBP constructs, only a high level of positive attitudes towards EBP was commonly present in those who showed high use of EBP over time. The qualitative findings suggest that personal and peer experiences, client preferences, and positive patient outcomes were key facilitators for EBP. Frequently encountered organizational barriers included time constraints, lack of access to databases, research opportunities, CPD activities, peer and financial support.

The decrease in the use of EBP could be explained by several individual and organizational factors. Findings from the current study and those from previous studies [57–61] converge to suggest that practitioners highly value their own personal practice experiences and those of their peers as a primary source of knowledge in making clinical decisions. With time and clinical acumen, practitioners' confidence in their abilities increases, and they appear to rely on their experiences as the only or primary source of evidence, especially when they have had successful past experiences in doing so [62, 63]. As experience increases, practitioners may need

to rely less on formal forms of evidence such as practice guidelines, as they may have already integrated and consolidated this evidence into their practice. If clinical acumen offsets the use of formal forms of evidence, then the lower use of EBP is unsurprising; however, this may be problematic as it could mean that practitioners would fail to apprise themselves of new scientific developments. What is the relationship between increasing experience and degree of being an "evidence-based" practitioner is a question that requires further exploration.

Practitioners also rely to a great extent on peer feedback and recommendations, because in their opinion, consulting peers is more convenient and effective compared to engaging in the traditional steps of the EBP process, including searching the literature. Peer recommendation is argued to be a vital and necessary source of evidence in complex clinical situations when literature fails to address practical aspects, specific contexts, or when evidence from randomized clinical trials conflicts with patient preferences [4, 64–66]. Acknowledging that peers are a part of a powerful social process of co-construction of knowledge [67, 68], a growing body of literature suggests that consulting one's peers promotes professional networking, which in turn positively influences the development of collective EBP competencies rather than targeting individual discrete skills [58, 69]. In an era of rapidly changing work contexts and continuously evolving scientific evidence, being involved in and learning from a community can enhance one's sense of belonging, promote transfer of learning to practice, and foster motivation among practitioners to develop their EBP competencies that can ultimately lead to a culture change [29, 70]. Despite the potential benefits of collaboration and peer consultation [57, 58], such practices could become challenging if targeted to a few select individuals who may, or may not, continuously monitor the evolution of evidence, or view EBP in a positive light. To mitigate for this, practitioners can develop broader professional networks and communities of practice for optimal development of EBP competencies that consider the value of peers in clinical decision-making [29, 58]. Reliance on one's peers has not been acknowledged as central to EBP in the extant literature. Indeed, the literature portrays EBP as a largely individualistic process. Therefore, a promising avenue for future research may be to further understand the merits and drawbacks of including peers as legitimate source of evidence in the core conceptualization of evidence-based decision-making.

Despite the gradual decrease in the use of EBP over time amongst two thirds of participants, those who started with high use of EBP after graduation continued to do so over time, and a subset showed improvement in the use of EBP. A potential reason for the sustained or improved use of EBP could be the satisfaction that practitioners experience when the scientific evidence is aligned with clients' needs and expectations. In line with this argument, more than half of our participants shared that their decisions to use evidence in practice are driven by the clients' safety and their preferences. Moreover, they are more likely to use evidence in practice when past experiences have resulted in positive patient outcomes. This is indeed a promising finding in that it offers valuable insights that can be used during formal graduate and CPD activities to positively influence practitioners' confidence and attitudes towards EBP, and ultimately, improve EBP behaviors. Importantly, it is also likely that these participants have favorable work contexts and access to resources such as databases, protected time, research and CPD activities. These affordances may have created the conditions for maintaining or improving EBP [5, 13, 15, 17, 71]. Organizational culture, leadership style, availability of financial, infrastructural, and human resources, and organizational mandate also play important role in influencing EBP [72, 73].

Although all constructs related to EBP are important, attitudes towards EBP play a key role in one's inclination to adopt EBP. Indeed, attitudes are strongly associated with practitioners' behaviors [74] and are a known precursor to practitioners' decision to adopt and apply evidence to their practice [75]. Our study shows that among all EBP constructs, only a high level

of positive attitudes towards EBP can help practitioners sustain a high use of EBP over time. This key finding could be taken as a call to revise formal entry level education curricula as well as design CPD training activities aimed at helping practitioners embrace the core ethos of EBP so that sustainable behavioral changes can occur [52]. These objectives can be achieved by: 1) increasing the relevance of the evidence to practice through hands-on training activities and real-life examples of EBP success (i.e., positive patient outcomes); 2) motivating practitioners to continuously stay up to date with the evidence so that they can provide optimal patient care in an already overburdened healthcare system; motivation can be enhanced though incentives and rewards that are both intrinsic (e.g., evidence of benefits for patients and their families;) and extrinsic (e.g., awards recognizing those who apply best practices); 3) validating practitioners' experiences and expertise as key to the EBP process.

Expectedly, time constraint was once again found to be a major barrier to EBP. There is certainly no shortage of research which suggests that EBP is a complex and time consuming process, and limited or no protected time significantly hinders practitioners' ability to systematically search and optimally implement research evidence in practice [3, 15, 28]. While acknowledging that practitioners are already overburdened with competing tasks and have limited time to look for evidence, they can still stay up to date with the evidence informed best practices by registering to email alerts, subscribing to evidence summaries, and being a part of communities of practice.

Finally, participants frequently experienced organizational barriers (e.g., lack of access to databases, research opportunities, CPD activities, peer and financial support, and unavailability of protected time) across all trajectories that might have negatively influenced their use of EBP [5, 27, 62, 76, 77]. Even 30 years after the inception of the EBP movement [78], there are serious and growing challenges in using evidence in practice that will persist unless systems-level changes are implemented to promote a sustainable culture of EBP [35, 36]. For instance, macro-level changes in an organization's paradigm and transformative leadership styles could promote a culture of continuous practice improvement and efficiency, which is an integral part of the EBP mandate. Moreover, an organizations' emphasis on reflective and collaborative practices whereby peer exchanges and co-construction are viewed as promising mechanisms to fostering EBP, can create a favorable climate that supports EBP agenda [3, 79–81]. These systems-level changes could contribute to restoring some of the lost agency that practitioners have reported in the literature [37]. With increased autonomy in the decision-making processes, practitioners may feel empowered to take control of their practice and make evidence-based decisions that are in the best interest of their patients, even in the face of resource constraints, variations in the nature of the available evidence and organizational pressures [82].

### Implications for research and practice

Though the decline in the use of EBP amongst two thirds of the trajectory participants may seem concerning for all sectors (e.g., education, practice, and policy), two potentially important issues need to be addressed as they may mitigate such concerns. First, the decline in the use of EBP (i.e., searching for evidence) may be associated with the limited availability of new and applicable clinical evidence in the field rather than be an actual decrease in the use of EBP [83]. This is particularly more likely in cases where the practitioners have already consolidated the available evidence into their decision-making process; searching for evidence would then seem to be a redundant exercise in such instances. As clinicians gain experience, they may show high use of EBP (as measured by the instrument in this study) only when faced with uncertainty or in light of a change in practice setting and/or clientele. Future research using experienced clinicians could be helpful in testing this hypothesis. Second, it remains unclear if,

and to what degree this decline poses serious and immediate threats to patients, organizations, and the overall healthcare system. For instance, how clinically meaningful are these declines? And do they warrant prompt attention and intervention? To address this question, we posit that there is an urgent need for concerted efforts, which will only be possible when key stakeholders (e.g., professional associations, educational institutions, practice settings, and professional regulatory bodies) work together. Currently, the education, professional practice and policy sectors are working in silos in the pursuit of the EBP agenda [10, 84]. This siloed approach will continue to act as deterrent and create uncertainty in practitioners regarding how to best apply evidence in practice [79]. Moreover, a rigorous investigation including stakeholders from all four sectors (education, research, policy, and practice) is essential to understand if and how the decline in EBP could pose challenges to the delivery of rehabilitation services in a rapidly evolving and highly complex healthcare system.

## Strengths and limitations

A major strength of this work is the use of multiple rigorous methodological and analytical approaches (i.e., group-based trajectory modeling and TDF guided content analyses) to answer our research questions. We also used a variety of procedures to ensure analytical rigor. For instance, we ensured credibility through debriefing and rapport building with the focus group participants. We ensured transparency, dependability, and reflexivity of data analyses and interpretation processes through multiple and iterative team debriefing discussions. Finally, the researchers had no relationship with the participants of this study as they were selected on a volunteer basis and worked in different practice settings.

These strengths notwithstanding, this study is not without limitations. First, the sample size was small. Due to the peak pandemic situation at the time of T1, T2 and T3 time points (i.e., years 2019–21), we received less than the anticipated number of survey responses and volunteers for the focus groups. Though the small sample size did not impact the choice of our analytical approach (group-based trajectory modeling is known to be useful in situations when sample sizes are relatively small), it did prevent us from performing cross comparative and correlational analyses, which is why we decided to provide means and standard deviations only. Second, the challenges associated with the pandemic forced us to change the platform for the focus groups from face-to-face to an online format (i.e., Zoom platform); this may have affected the richness of the discussions. Lastly, although we closely followed the established guidelines while recording, analyzing, interpreting, and reporting our study, the findings might have limited transferability beyond both Canadian and rehabilitation sciences contexts.

## Conclusions

In this longitudinal study, we found four trajectories on the use of EBP, of which two-third showed a decline in use over time. Among all EBP constructs, positive attitudes towards EBP seem to be necessary to ensure high use of EBP. High EB activities, knowledge, confidence, and resources do not appear to be sufficient to prevent the EBP decline. Personal and peer experiences, client needs and expectations, and availability of resources were perceived to influence EBP the most. The findings of this study might support professional OT and PT programs in Canada to review and revise their curricula, and to inform the knowledge translation interventions designed to positively influence the attitudes and behaviors of practitioners towards EBP. Moreover, it may be worthwhile for a team such as ours to meet with stakeholders from education, practice, and policy sectors to discuss the implications of our findings and

design next steps on how we can collaboratively contribute towards bridging the research-practice gap in both professions.

## Supporting information

**S1 Appendix. COREQ criteria used for reporting qualitative phase.**
(DOCX)

**S2 Appendix. Guide for focus group discussions.**
(DOCX)

**S3 Appendix. Time points wise content analysis guided by Theoretical Domain Framework (TDF).**
(DOCX)

**S4 Appendix. Summary of OTs and PTs specific beliefs and sample quotes assigned to the relevant domains.**
(DOCX)

## Author Contributions

**Conceptualization:** Annie Rochette, Nancy E. Mayo, Marie-France Valois, André E. Bussières, Richard Debigaré, Lori Jean Letts, Joy C. MacDermid, Tatiana Ogourtsova, Helene J. Polatajko, Susan Rappolt, Nancy M. Salbach, Aliki Thomas.

**Data curation:** Annie Rochette, Lori Jean Letts, Aliki Thomas.

**Formal analysis:** Muhammad Zafar Iqbal, Annie Rochette, Nancy E. Mayo, Marie-France Valois, André E. Bussières, Tatiana Ogourtsova, Aliki Thomas.

**Funding acquisition:** Aliki Thomas.

**Investigation:** Sara Ahmed, Aliki Thomas.

**Methodology:** Annie Rochette, Nancy E. Mayo, Marie-France Valois, Aliki Thomas.

**Project administration:** Annie Rochette, Aliki Thomas.

**Resources:** Aliki Thomas.

**Software:** Marie-France Valois.

**Supervision:** Aliki Thomas.

**Validation:** Muhammad Zafar Iqbal, Annie Rochette, Nancy E. Mayo, André E. Bussières, Sara Ahmed, Richard Debigaré, Lori Jean Letts, Joy C. MacDermid, Tatiana Ogourtsova, Helene J. Polatajko, Susan Rappolt, Nancy M. Salbach.

**Visualization:** Marie-France Valois.

**Writing – original draft:** Muhammad Zafar Iqbal, Annie Rochette, Aliki Thomas.

**Writing – review & editing:** Muhammad Zafar Iqbal, Nancy E. Mayo, Marie-France Valois, André E. Bussières, Sara Ahmed, Richard Debigaré, Lori Jean Letts, Joy C. MacDermid, Tatiana Ogourtsova, Helene J. Polatajko, Susan Rappolt, Nancy M. Salbach, Aliki Thomas.

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
