## [Decision Letter · Decision Letter 0]

2 Dec 2022

PONE-D-22-19314Exploring how evidence-based practice of occupational and physical therapists evolves: A longitudinal mixed methods national studyPLOS ONE

Dear Dr. Thomas,

Thank you for submitting your manuscript to PLOS ONE. I do want to personally apologise for the delay in returning the results of the review as I have a very difficult time securing two reviewers. After careful consideration, we feel that it has merit but does not fully meet PLOS ONE’s publication criteria as it currently stands. Therefore, we invite you to submit a revised version of the manuscript that addresses the points raised during the review process.

The topic of your paper is potentially valuable and important. However, both reviewers made important suggestions that will strengthen the paper markedly. I concur with their suggestions. Consider all their comments but pay particular attention to the suggestion of a further discussion of the findings and limitations of the methodology especially the impact of participants' characteristics.

We look forward to receiving your revised manuscript.

Kind regards,

Yu-Wei Ryan Chen, PhD

Academic Editor

PLOS ONE

https://journals.plos.org/plosone/s/fileid=ba62/PLOSOne_formatting_sample_title_authors_affiliations.pdf.

3. PLOS requires an ORCID iD for the corresponding author in Editorial Manager on papers submitted after December 6th, 2016. Please ensure that you have an ORCID iD and that it is validated in Editorial Manager. To do this, go to ‘Update my Information’ (in the upper left-hand corner of the main menu), and click on the Fetch/Validate link next to the ORCID field. This will take you to the ORCID site and allow you to create a new iD or authenticate a pre-existing iD in Editorial Manager. Please see the following video for instructions on linking an ORCID iD to your Editorial Manager account: https://www.youtube.com/watch?v=_xcclfuvtxQ.

Reviewers' comments:

Reviewer's Responses to Questions

**Comments to the Author**

1. Is the manuscript technically sound, and do the data support the conclusions?

Reviewer #1: Partly

Reviewer #2: Yes

2. Has the statistical analysis been performed appropriately and rigorously? 

Reviewer #1: Yes

Reviewer #2: I Don't Know

3. Have the authors made all data underlying the findings in their manuscript fully available?

Reviewer #1: Yes

Reviewer #2: Yes

4. Is the manuscript presented in an intelligible fashion and written in standard English?

Reviewer #1: Yes

Reviewer #2: Yes

5. Review Comments to the Author

Reviewer #1: Overall, the paper is well-conducted, and answers a question that does not seem to be assessed elsewhere.

Research aim is clearly formulated, and intent of analysis is clear, and relevant methods have been implemented. Statistical models are sufficient, and conclusions drawn are in line with the data.

However, the argumentation for the justification of the study is limited by the following points. It is unclear what the value of this study is given the context of the evidence. It is clearly argued that research has shown limited implementation of EBP in multiple settings in OT and PT populations, but it is unclear why it is relevant to assess this tendency in early career clinicians, and what can be gained from the study for end-users? This point is made clearer in the discussion but is taken implicitly in the introduction and should be added.

Methodologically the study is well described and seem well-conducted for the chosen methods. The proportion of participants is skewed between EBP groups, and the number of participants decreased across time-points. How was the influence of low sample size at different time-points assessed, and how could this influence interpretation of results?

Concerning qualitative methods, the researchers’ characteristics that may influence the research, including personal qualifications/experience, relationship with participants, assumptions, and presuppositions were not reported in the methods or possible influences were not discussed. Could any of these factors have affected study results?

What criteria was used to decide that no further sampling was required, or were all volunteers included in the study? Which influence did this have on interpretation of results?

It appear that there is a spread in the characteristics of the participants. Was this distribution or characteristics of individual participants known before or during the Focus group interviews, and what measures did authors take to not make this knowledge influence inferences or was knowledge of participant characteristics included in interpretation of respondent answers? Methods used by authors appear to remedy this, but the rationale for how authors attempted to ensure that the data is representative of the respondents is not made explicit. Authors state that they probed for specific examples from clinical practice, were conclusions drawn from coding of these examples checked with the participants themselves? How was checking of results conducted?

Authors do not discuss the strengths and limitations of the study findings in the discussion, and it is unclear how the specific context of the data-collection influence interpretation and transferability of findings.

Concerning the “Moving forward” section. The aim to assess whether the changes are relevant should be part of the justification of the study. Why is this relevant if it is unclear that it will lead to usable results of relevance to end-users? And how can the data generated in the study inform future studies?

2: Title: ”how EBP… evolves”. Evolves from what to where, for whom? “Exploring how evidence-based practice of early career occupational and physical therapists evolves in the first three years of practice: A longitudinal mixed methods national study”.

L. 247. Table 1. It is unclear whether the reported number of respondents at T0-T4 include non-respondents. Respondents to the questions of “Current clinical setting” sums to 147 with 257 respondents reported, with only 3 missing. This is the same across all categories and time-points. The “missing” category should represent the number of non-responses, as defined by asterisk.

l. 230-234: It is unclear what is meant by “relevant” given the 3 listed criteria. Were specific domains listed with most to least relevance to ex. the presence of conflicting beliefs for the students themselves or in relation to the study aim? Was this assessment conducted in the context of the others analyses?

l. 421-423: Unclear what is meant by limited evidence in the field. In the research field on practices on EBP-utilization in practice (i.e. lacking knowledge on how practitioners implement EBP) or specific to the field of clinicians (ex. lacking evidence to implement in practice? Or similar?). And does “availability” refer to structural limitations ex. journal access? – It is unclear how this argument is relevant to the aim of the study (early career researchers in the first 3 years)?

l.423-428: This is argument does not propose or support recommendations for future clinical practice or research.

l.428-437: The second argument in ”moving forward” state possible limitations of a current system, but does not utilize results of study or possible interpretations of the evidence to provide knowledge on how to move past these problem for clinicians, or specific recommendations for future studies. Which concerted efforts do authors propose?

Reviewer #2: Thank you for the opportunity to review this interesting and ambitious manuscript, which should be of great interest to researchers and practitioners in the field. Please see attached file which contains my comments on the manuscript.

Best wishes.

6. PLOS authors have the option to publish the peer review history of their article (what does this mean?). If published, this will include your full peer review and any attached files.

Reviewer #1: No

Reviewer #2: No

---

## [Author Response · Author response to Decision Letter 0]

17 Feb 2023

Please note that the page and line numbers mentioned below are according to the "clean version" of the revised paper. Red colored text in the manuscript represents revised and/or newly added content. Specific changes can be viewed in the tracked file.

RESPONSE TO THE EDITOR:

Editor: Journal requirements:

https://journals.plos.org/plosone/s/fileid=ba62/PLOSOne_formatting_sample_title_authors_affiliations.pdf.

Authors: We have gone through the guidelines again and made sure that the paper is formatted as per journal guidelines. If there is anything specific that we missed, please let us know and we will address it promptly.

Editor: 2. We note that you have indicated that data from this study are available upon request. PLOS only allows data to be available upon request if there are legal or ethical restrictions on sharing data publicly. For more information on unacceptable data access restrictions, please see http://journals.plos.org/plosone/s/data-availability#loc-unacceptable-data-access-restrictions.

Authors: We did not obtain permission from the institutional review board for public availability of the data, nor did we obtain consent from participants. Therefore, we cannot share or make the data public. However, the data are available upon request for verification purposes. For confirmation of the reported limitations, the editorial office may contact the institutional review board of McGill University at submit2irb.med@mcgill.ca. The reference number of IRB letter is A10-B55-16B.

Editor: 3. PLOS requires an ORCID iD for the corresponding author in Editorial Manager on papers submitted after December 6th, 2016. Please ensure that you have an ORCID iD and that it is validated in Editorial Manager. To do this, go to ‘Update my Information’ (in the upper left-hand corner of the main menu), and click on the Fetch/Validate link next to the ORCID field. This will take you to the ORCID site and allow you to create a new iD or authenticate a pre-existing iD in Editorial Manager. Please see the following video for instructions on linking an ORCID iD to your Editorial Manager account: https://www.youtube.com/watch?v=_xcclfuvtxQ

Authors: The corresponding author’s ORCID ID has been updated in the submission portal. Below are available ORCID IDs of other authors. 

- Muhammad Zafar Iqbal: https://orcid.org/0000-0002-5605-8143

- Annie Rochette: https://orcid.org/0000-0002-0189-0987

- André E. Bussières: https://orcid.org/0000-0002-2818-6949

- Nancy Salbach: https://orcid.org/0000-0002-6178-0691

- Aliki Thomas: https://orcid.org/0000-0001-9807-6609

RESPONSE TO REVIEWER 1:

Reviewer 1: -Overall, the paper is well-conducted, and answers a question that does not seem to be assessed elsewhere.

Research aim is clearly formulated, and intent of analysis is clear, and relevant methods have been implemented. Statistical models are sufficient, and conclusions drawn are in line with the data. However, the argumentation for the justification of the study is limited by the following points

Authors: Thank you for these comments. 

We have now addressed each point below.

Reviewer 1: - It is unclear what the value of this study is given the context of the evidence. It is clearly argued that research has shown limited implementation of EBP in multiple settings in OT and PT populations, but it is unclear why it is relevant to assess this tendency in early career clinicians, and what can be gained from the study for end-users? This point is made clearer in the discussion but is taken implicitly in the introduction and should be added.

Authors: We have now made the following changes in the Introduction section:

- We explicitly focused on early career OTs and PTs in the revised Introduction. 

- We added a revised section at the end of the Introduction that highlights why it is important to assess EBP in early career clinicians, and how the results from this study may benefit end-users. The newly added section reads as follows:

Most of these studies have been conducted at a specific time point or with experienced practitioners; the data from these studies cannot be used to predict the transition of EBP among the early career practitioners, all of whom were highly trained in EBP. Moreover, most previous studies were either small scale, single site studies or lacked robust analytical approaches and measures [17]. 

There was, therefore, a need for a longitudinal nationwide study to track EBP and its associated factors among early career OT and PT practitioners. Longitudinally examining if and how EBP changes over time, and which factors influence this change might support curricular reforms of professional OT and PT programs across Canada. This exploration may also inform future knowledge translation interventions designed to positively influence EBP competencies. 

Changes can be seen on Page 7, Lines 132-141

Reviewer 1: - Methodologically the study is well described and seem well-conducted for the chosen methods. The proportion of participants is skewed between EBP groups, and the number of participants decreased across time-points. 

- How was the influence of low sample size at different time-points assessed

- And how could this influence interpretation of results?

Authors: - Thank you for the comment. The COVID-19 pandemic was at its peak during time points 2, 3 and 4, which significantly impacted data collection. Keeping this limitation in mind and after discussion with our team, we decided to provide the descriptive data (as presented in Table 1) and avoided drawing any statistical comparisons between variables/ participant characteristics. We have highlighted this limitation in the newly added Limitations section on Page 24. The new limitations section reads as follows:

These strengths notwithstanding, this study is not without limitations. First, the sample size was small. Due to the peak pandemic situation at the time of T1, T2 and T3 time points (i.e., years 2019-21), we received less than the anticipated number of survey responses and volunteers for the focus groups. Though the small sample size did not impact the choice of our analytical approach (group-based trajectory modeling is known to be useful in situations when sample sizes are relatively small), it did prevent us from performing cross comparative and correlational analyses, which is why we decided to provide means and standard deviations only. Second, the challenges associated with the pandemic forced us to change the platform for the focus groups from face-to-face to an online format (i.e., Zoom platform); this may have affected the richness of the discussions. Lastly, although we closely followed the established guidelines while recording, analyzing, interpreting, and reporting our study, the findings might have limited transferability beyond both Canadian and rehabilitation sciences contexts.

- We do wish to highlight, however, that the smaller sample size did not affect our main objective – that is – to measure the change in EBP over time through group-based trajectory modeling. GBTM only shows the trend of a construct over time and is not influenced by the sample size. In fact, as discussed in the literature (Loughran & Nagin, 2006; Nagin & Nagin, 2005), GBTM is a promising strategy in cases where sample sizes are small.

Changes can be seen on Page 24, lines 480-491

Reviewer 1: - Concerning qualitative methods, the researchers’ characteristics that may influence the research, including personal qualifications/experience, relationship with participants, assumptions, and presuppositions were not reported in the methods or possible influences were not discussed. Could any of these factors have affected study results?

Authors: Personal qualification and experience: Three senior authors (AT, AR and AB) involved in designing and conducting the qualitative phase of this study, are seasoned EBP, knowledge translation and implementation science researchers with over two decades of research experience and established research programs. They have all used the TDF in the past and these studies have already been published in peer reviewed international journals (Bussieres et al., 2012; Bussières et al., 2015; Rochette et al., 2020). This reflects their experience with the framework and study design. We have explained this in S1 Appendix.

Relationship with participants: the researchers had no relationship with the study participants. Interviewees were selected on a volunteer basis, and they worked in different practice settings. We consider this a strength of our study and therefore mentioned it in the newly added Strengths section on Page 23.

Assumptions and presuppositions: We closely followed the COREQ qualitative research guidelines (see S1 Appendix). One experienced qualitative researcher (TO) coded and four researchers (MZI, AB, AT, AR) reviewed the data independently and then came together to discuss the findings and develop agreement on the assignment of utterances into their relevant domains. TO was not involved in any stage of data collection, which is why we do not foresee any bias in the interpretation of the results. Moreover, the findings and interpretations were also shared with the full research team and feedback was obtained to ensure transparency. 

In S1 Appendix, we have now explained how we followed all steps to ensure transparency. 

Changes can be seen on Page 23, lines 472-479 and S1 Appendix

Reviewer 1: - What criteria was used to decide that no further sampling was required, or were all volunteers included in the study? 

- Which influence did this have on interpretation of results?

Authors: We included all those who volunteered to participate in the study. Due to the peak pandemic situation at the time of the qualitative data collection (i.e., years 2019-21), we received less than the anticipated number of volunteers for this phase of the study. The pandemic also pushed us to change the platform of the FGDs from face-to-face to an online format (i.e., we used the Zoom platform). We have highlighted this in the newly added limitations section on Page 24. The new limitations section reads as follows:

These strengths notwithstanding, this study is not without limitations. First, the sample size was small. Due to the peak pandemic situation at the time of T1, T2 and T3 time points (i.e., years 2019-21), we received less than the anticipated number of survey responses and volunteers for the focus groups. Though the small sample size did not impact the choice of our analytical approach (group-based trajectory modeling is known to be useful in situations when sample sizes are relatively small), it did prevent us from performing cross comparative and correlational analyses, which is why we decided to provide means and standard deviations only. Second, the challenges associated with the pandemic forced us to change the platform for the focus groups from face-to-face to an online format (i.e., Zoom platform); this may have affected the richness of the discussions. Lastly, although we closely followed the established guidelines while recording, analyzing, interpreting, and reporting our study, the findings might have limited transferability beyond both Canadian and rehabilitation sciences contexts.

Changes can be seen on Page 24, lines 480-491

Reviewer 1: -It appear that there is a spread in the characteristics of the participants. 

- Was this distribution or characteristics of individual participants known before or during the Focus group interviews, and 

- what measures did authors take to not make this knowledge influence inferences or was knowledge of participant characteristics included in interpretation of respondent answers? 

Authors: As mentioned on Page 10, we used purposive sampling to recruit practicing OTs and PTs that had participated in the quantitative survey. We did not use any other inclusion criteria (i.e., demographic characteristics, qualification, practice context), which is why we do not anticipate any influence of these characteristics on the data interpretation. The one inclusion criterion was having completed the survey. The main objective of our FGDs was to delve deeper into our quantitative findings and not to explore the relationship between participants’ use of EBP and their characteristics. We have now clarified this under the Study Design section on page 8. The newly added content reads as follows:

In our case, the quantitative phase was the dominant component as it led to the identification of the EBP trajectories (primary study objective), whereas the qualitative phase was sequentially (after each survey data collection time) integrated throughout the duration of the study to deepen our understanding of the individual and organizational facilitators and barriers to EBP.

Page 8, lines 157-161

Reviewer 1: Methods used by authors appear to remedy this, but the rationale for how authors attempted to ensure that the data is representative of the respondents is not made explicit. 

- Authors state that they probed for specific examples from clinical practice, were conclusions drawn from coding of these examples checked with the participants themselves? 

- How was checking of results conducted?

Authors: We did not perform member checking of the qualitative findings. Birt et al. (2016) argue, and we concur with their view that the conventionally used member checking technique might not add value to the findings nor to its juxtaposition with the interpretative stance of qualitative research. This is because member checking might be confounded by epistemological and methodological challenges that include: the changing nature of interpretations of phenomena over time, the ethical issue of returning data to participants, the dilemma of anticipating and assimilating the disconfirming voices, and deciding who has ultimate responsibility for the overall interpretation. 

To maximize the trustworthiness of the findings, four qualitative research experts (MZI, AB, AT, AR) independently reviewed the codes and utterances that were generated from the data and agreed on the findings. This is stated on Page 10 under data analysis section. Please also see S1 Appendix.

Reviewer 1: -Authors do not discuss the limitations of the study findings in the discussion, and it is unclear how the specific context of the data-collection influence interpretation and transferability of findings.

Authors: In our newly added limitations section on Page 24, we have discussed this point. It reads as follows:

Lastly, although we closely followed the established guidelines while recording, analyzing, interpreting, and reporting our study, the findings might have limited transferability beyond both Canadian and rehabilitation sciences contexts.

Page 24, lines 489-491

Reviewer 1: -Concerning the “Moving forward” section. The aim to assess whether the changes are relevant should be part of the justification of the study. 

- Why is this relevant if it is unclear that it will lead to usable results of relevance to end-users? 

- And how can the data generated in the study inform future studies?

Authors: We have now explicitly highlighted the value of this study for the end users in the Introduction section on Page 7. It reads as follows:

Most of these studies have been conducted at a specific time point or with experienced practitioners; the data from these studies cannot be used to predict the transition of EBP among the early career practitioners, all of whom were highly trained in EBP. Moreover, most previous studies were either small scale, single site studies or lacked robust analytical approaches and measures [17]. 

There was, therefore, a need for a longitudinal nationwide study to track EBP and its associated factors among early career OT and PT practitioners. Longitudinally examining if and how EBP changes over time, and which factors influence this change might support curricular reforms of professional OT and PT programs across Canada. This exploration may also inform future knowledge translation interventions designed to positively influence EBP competencies.

How the study findings can inform future research are now included in the revised section “Implications for research and practice” on Pages 23-24. This is in addition to the recommendations for research that have already been highlighted in the Discussion section.

Page 7, Lines 132-141 and Pages 23-24, lines 450-452, 457-458, 467-470.

Reviewer 1: -2: Title: ”how EBP… evolves”. Evolves from what to where, for whom? “Exploring how evidence-based practice of early career occupational and physical therapists evolves in the first three years of practice: A longitudinal mixed methods national study”.

Authors: We have now rephrased the title of the paper. The new title is: Exploring if and how evidence-based practice of occupational and physical therapists evolves over time: A longitudinal mixed methods national study

Page 3, lines 38-39

Reviewer 1: -L. 247. Table 1. It is unclear whether the reported number of respondents at T0-T4 include non-respondents. Respondents to the questions of “Current clinical setting” sums to 147 with 257 respondents reported, with only 3 missing. This is the same across all categories and time-points. The “missing” category should represent the number of non-responses, as defined by asterisk.

Authors: All questions related to employment were only answered by those working at the time of the survey. So, time point 0 was within the first 6 weeks from graduation from their respective OT /PT program. This means that 110 out of 257 were not working at the time of the survey, which is why their responses were not included. We have now clarified it in Table 1 (variables and footnote sections). 

Pages 13-14, Table 1 and Page 14, line 273 (footnote)

Reviewer 1: -l. 230-234: It is unclear what is meant by “relevant” given the 3 listed criteria. 

- Were specific domains listed with most to least relevance to ex. the presence of conflicting beliefs for the students themselves or in relation to the study aim?

- Was this assessment conducted in the context of the others analyses?

Authors: - By relevant we mean the relevance of beliefs related to the use of EBP in practice and not to the aim of the study; by conflict we mean the conflicting beliefs in participant responses. The categorization of beliefs and utterances according to TDF domains is a commonly used method in TDF based studies (To et al., 2022). 

- No, the assessment was not conducted in the context of the other analyses; the TDF analyses were independent of the quantitative analyses.

Reviewer 1: -l. 421-423: Unclear what is meant by limited evidence in the field. 

- In the research field on practices on EBP-utilization in practice (i.e. lacking knowledge on how practitioners implement EBP) or specific to the field of clinicians (ex. lacking evidence to implement in practice? Or similar?). 

- And does “availability” refer to structural limitations ex. journal access? 

Authors: We have now rephrased this sentence on Page 22 to improve clarity. The revised line reads as follows:

First, the decline in the use of EBP (i.e., searching for evidence) may be associated with the limited availability of new and applicable clinical evidence in the field rather than be an actual decrease in the use of EBP [86].

Page 22, lines 450-452

Reviewer 1: -l.423-428: This is argument does not propose or support recommendations for future clinical practice or research.

Authors: We have now rephrased the heading from “Moving forward” to “Implications for theory and practice”. 

Page 22, line 447

Reviewer 1: -l.428-437: The second argument in ”moving forward” state possible limitations of a current system, but does not utilize results of study or possible interpretations of the evidence to provide knowledge on how to move past these problem for clinicians, or specific recommendations for future studies. Which concerted efforts do authors propose?

Authors: Same comment as above. 

RESPONSE TO REVIEWER 2:

Reviewer 2: Thank you for the opportunity to review this interesting and ambitious manuscript, which should be of great interest to researchers and practitioners in the field. Please see attached file which contains my comments on the manuscript.

Authors: Thank you for the kind comments. We have now addressed each point below.

Reviewer 2: Abstract

-I recommend fewer abbreviations in the abstract. For example, EB, FGD, GBTM could be written out (the sentence in Results that begins with GBTM could be reworded). Abbreviations in the beginning of a sentence is against most writing guidelines.

Authors: We removed unnecessary abbreviations from the Abstract. 

Pages 3-4

Reviewer 2: -The study aim is not formulated the same as in the main text.

Authors: We revised the study aim in the Abstract section to match with the one that is found after the introduction section on Page 3. The revised study aim reads as follows:

The aim of the study was to measure and understand how EBP evolves over the first three years after graduation among Canadian OTs and PTs, and how individual and organizational factors impact the continuous use of EBP.

Page 3, lines 46-48

Reviewer 2: -Please state the response rate for the survey. Maybe also the retention rates.

Authors: We added the response and retention rates in the results section of the abstract. The revised Results section reads as follows:

Of 1700 graduates in 2016-2017, 257 (response rate=15%) responded at baseline (T0) (i.e., at graduation), and 83 (retention rate=32%), 75 (retention rate=29%), and 74 (retention rate=29%) participated at time point 1 (T1: one year into practice), time point 2 (T2: two years into practice, and time point 3 (T3: three years into practice) respectively. Group-based trajectory modeling showed four unique group trajectories for the use of EBP.

Page 3, lines 57-61

Reviewer 2: Background

-Could you provide more examples of EB activities? Simply ”informal information sharing…” seems quite limited, surely there must be other examples of other types of activities?

Authors: We have now added more examples of use of EBP and EB activities on Page 5. The revised section reads as follows:

In our previous work, use of EBP was defined as “the actual application of EBP concepts, tools, and procedures into specific actions” [5] (p.3). Identifying a gap in knowledge related to a patient situation, effectively conducting an online literature search to address the research question, and critically appraising the strengths and weaknesses of study methods are some examples of use of EBP. Evidence-based (EB) activities refers to “the implementation of research evidence to the surrounding environment” [5] (p.3). Informally sharing and discussing literature/research findings with colleagues or patients, integrating research evidence with expertise, and making time and reading research reports are examples of EB activities.

Page 5, lines 84-92

Reviewer 2: -Please avoid using abbreviations in the beginning of a sentence (EB, but other abbreviations elsewhere in the manuscript).

Authors: We corrected this throughout the manuscript. 

Reviewer 2: -Not all references 6-9 seem to deal with EBP in rehabilitation services. Please review and, if appropriate, select more relevant papers to support the statement.

Authors: We updated the references # 8 and 9 with rehabilitation specific literature. The news added references are as follows:

8. Dijkers MP, Murphy SL, Krellman J. Evidence-based practice for rehabilitation professionals: concepts and controversies. Arch Phys Med Rehabil. 2012;93: S164–S76. 

9. Jutai JW, Teasell RW. The necessity and limitations of evidence-based practice in stroke rehabilitation. Top Stroke Rehabil. 2003;10: 71–8.

Pages 8-9, lines 527-530

Reviewer 2: -The paragraph on the top half of page 6 could be written more concisely. For example, author names do not need to be stated; what the study did does not need to be stated, only their findings relative to the point you wish to make. Several of the cited papers say more or less the same so could be merged together.

Authors: We revised this paragraph to make it more concise. 

It now reads as follows:

Despite an increased emphasis on EBP reflected in major changes in professional curricula in Canada and elsewhere in the world, the use of EBP by OT and PT practitioners remains a challenge [13–15]. This is somewhat concerning when studies show that early career practitioners generally hold positive attitudes towards EBP [14,16,17], but that only half use EBP [13]. Findings from our nationwide study showed that two-thirds of the participating OT and PT graduates reported using EBP upon entry to practice [5]. In studies including different health professions, PTs showed more positive attitudes towards EBP but lesser use of EBP than other professions such as physicians, nurses, podiatry and radiology [18,19]. Similarly, a moderate use of EBP was found in a survey of more than 1500 OTs in New Zealand [20]. In parallel, studies of engagement of practitioners in EB activities report inconsistent findings, ranging from reasonable [5,13,21] to suboptimal [20,22] involvement in such activities.

Page 6, lines 103-113

Reviewer 2: Methods

Study design and setting

-Although this heading says so, the study setting is not described here.

Authors: Since it was a national survey, we removed the word “Settings” from the heading. 

Page 8, line 148

Reviewer 2: -Please state what reporting guidelines you’ve used (e.g. STROBE, COREQ), and make sure you’ve followed them in your reporting. There are currently a few aspects, particularly in the Procedure sections that would benefit from your providing more details. This would certainly also increase transparency and credibility of the work and findings, especially the qualitative part.

Authors: We have now explicitly described how we used COREQ guidelines in data collection, analysis and interpretation. Please see S1 Appendix. 

S1 Appendix

Reviewer 2: -Please elaborate on the chosen type of Mixed methods and describe your integration strategies, i.e., how and where in the study process you’ve integrated the quantitative and qualitative data, whether the two data types were equally weighted/prioritised, etc. 

Consider moving the first three lines under Phase 2, qualitative to the design section, possibly also adding a line about Phase 1 as an introduction to the phase 2 part. This is important to improve clarity concerning your study design and type of Mixed methods used.

Authors: We have now completely revised the Study Design section and added more details for elaboration purpose. The revised study design section now reads as follows:

We conducted a longitudinal, cohort-based, mixed methods sequential explanatory study [40,41] spanning a period of three years (2016-17 until 2020-21). A sequential explanatory mixed methods study design, grounded in a postpositivist paradigm, is a methodology for sequentially collecting, analyzing, and interpreting the quantitative and qualitative data in a single study to synergistically investigate the same underlying phenomenon or research question [40,41]. More specifically, we used a fully mixed sequential dominant status design that mixes quantitative and qualitative research within one, or across different stages of the research process, but one component (either quantitative or qualitative) remains dominant and leads to the design of the other component for further exploration [42]. In our case, the quantitative phase was the dominant component as it led to the identification of the EBP trajectories (primary study objective), whereas the qualitative phase was sequentially (after each survey data collection time) integrated throughout the duration of the study to deepen our understanding of the individual and organizational facilitators and barriers to EBP.

- We did not consider data of both phases equally; the quantitative phase was the dominant phase of our study for the reasons explained above. We have made this explicit in the revised manuscript as well. We analyzed and interpreted the results of both phases separately as it was the best approach considering the research questions. However, we integrated the key findings of both phases in the discussion section to answer our research question. 

We have now moved the lines up in the Study Design section as advised.

Page 8, lines 149-161

Reviewer 2: -Please add ”explanatory” before sequential, as you have done in the abstract, and in line with the classification typology of MM designs.

Authors: We added the missing word “explanatory” in the study design section. Re: specific typology, we have now explained it in detail in the revised Study Design section. Please also see comment above. 

Page 8, lines 149-161

Reviewer 2: Please consider whether you could present the results in a more integrated manner. Presently, both analyses and results are presented as separate entities rather than integrated - so what does the integration consist of? This could be clarified. 

Authors: We understand the reviewer’s suggestion to integrate both sections as we also wrestled with the best way to present our findings; but considering the complexity of our study design, we decided to explain the quantitative and qualitative results separately for nuance and clarity, and then discuss them as a whole in the discussion. We would therefore prefer to keep this format.

Reviewer 2: Procedure

-The section would benefit from some more details. For example, where were the FGDs held, how long were the sessions, who moderated, etc. Please follow a reporting guideline for qualitative studies, e.g., COREQ (all items may not be applicable). Please consider providing the discussion guide in an appendix (see below regarding terminology).

Authors: We have now explained all steps in greater detail in S1 Appendix. We have now also provided the discussion guide as S2 Appendix. 

Please see S1 and S2 Appendices.

Reviewer 2: -Please consider presenting the 14 TDF domains, in text, table or figure.

Authors: We have now added the TDF domains in the text on Page 11. The newly added text reads as follows:

It includes 14 domains: knowledge, beliefs about capabilities, behavioral regulation, skills, beliefs about consequences, environmental context and resources, social influences, social/professional role and identity, emotions, goals, decision processes, reinforcement, optimism, and intention [53].

Page 11, lines 221-224

Reviewer 2: -Please add the original reference for the TDF (Michie 2005). (I have no relation to the author).

Authors: The original framework was published in 2005 (Michie et al., 2005) and further refined and validated in 2012 (Cane et al., 2012). In our study, we used the most recent and validated framework, which includes 14 domains instead of the original 12 domains in the 2005 version. The refined framework has a strengthened empirical base and provides a more rigorous method for theoretically assessing implementation problems, as well as professional and other health-related behaviors. Therefore, we would like to keep the current reference as the original source. 

Reviewer 2: Data analysis

-There is no mention of the analytical method/approach used. Content analysis is mentioned in Results and in the abstract, but not in methods. Please also specify whether the content analysis was quantitative or qualitative. It seems from the numbers presented in the results and App 1 (utterances, beliefs, etc) that it was mainly quantitative. Usually, in qualitative research, the richness of the data is more important than the quantity of statements or of participants who contributed to a finding. Interestingly, the guide you have used (Ref 54) states: ”Frequency count of belief statements is not warranted in the case of focus groups”.

Authors: We have now mentioned content analysis in the data analysis section of qualitative phase. 

We agree that the value of the qualitative data is in the exploration and not in quantification. However, following the style of previous studies that have used TDF (McGowan et al., 2020; Roberts et al., 2015; To et al., 2022), we have provided both quantitative and qualitative findings in S1 and S2 Appendices. 

Page 11, lines 233-234

Reviewer 2: Ethics

-was consent written or oral? Was it preceded by information about the study and its objectives?

-Please remove ”the” before ”ethical approval”.

Authors: We received written consent from the participants. In the consent form, we explained the study background, rationale and objectives in detail. A sample of consent form is available on request. 

Reviewer 2: Results

-Please state the number of eligble participants who received the survey at baseline and response rate.

Authors: On Page 8, we have already provided this information. The retention rate is given on Page 13, lines 258-260.

Reviewer 2: -Please add ”The” before ”majority”

Authors: We made this change. 

Page 13, line 260

Reviewer 2: -P. 14, line 264. The words ”more specifically, the posterior probability was” could be replaced by a semicolon after constructs. (improves conciseness) 

Authors: We made this change.

Page 15, lines 283-286

Reviewer 2: -P-16, line 305. Focus group discussions are typically just that – discussions rather than interviews. Please consider avoiding terms such as interviewed, interviewee. 

Authors: We replaced the word “interviewee” with “FGD participants” throughout the paper.

Reviewer 2: -P. 16, line 301-311. How did you determine which groups the participants belonged to?

Authors: FGD participants were the ones who participated in our annual survey and volunteered to participate in the follow up FGDs; therefore, we already had their demographic characteristics from our quantitative data. 

Reviewer 2: -Please consider expanding the qualitative findings section to summarize the key findings, including the most important individual and organizational factors that was perceived to influence their EBP use. There are several findings discussed in the Discussion section (e.g. own and peers’ experience, clients’ safety and preferences, positive patient outcomes, positive attitudes, organizational barriers, etc) that are not presented in the Results section. 

Authors: We discussed the key findings from the qualitative data in the Discussion section. Given the volume of data, we chose to report these as appendices. More detailed results from qualitative data can be found in S1 and S2 Appendices. We therefore believe that it would be redundant to present the same information in the text.

Reviewer 2:-Please consider clarifying ”constructs” by adding e.g. EBP or EBP-related before the word construct – in text as well as table heading and figure legend.

Authors: We replaced the word “constructs” with “EBP constructs” throughout the manuscript. 

Reviewer 2: Discussion

-Please begin the discussion by summarizing key findings from the quant and qual data, including the main individual and organizational factors that influence EBP use (i.e., your research question). More or less like you have done in the conclusion, line 439-444.

A statement in line with: ”The findings indicate that x, y and z are important factors…” would be an appropriate way to summarize what you found in answer to your second research question.

Authors: Thank you for this important point. We have now added new content at the start of the Discussion section to highlight the key quant and qual findings of our study. The new addition reads as follows:

In this longitudinal, mixed-method study that spanned a period of three years, we sought to explore if and how EBP evolves among OT and PT graduates entering practice and what factors are associated with their use of EBP over time. There are four key findings from the quantitative data: (1) a slight but steady decrease in the use of EBP amongst two thirds of the trajectory participants; (2) those who started with high use of EBP after graduation continued to do so over time; (3) only a small subset showed an increase in the use of EBP; and (4) among all EBP constructs, only a high level of positive attitudes towards EBP was commonly present in those who showed high use of EBP over time. The qualitative findings suggest that personal and peer experiences, client preferences, and positive patient outcomes were key facilitators for EBP. Frequently encountered organizational barriers included time constraints, lack of access to databases, research opportunities, CPD activities, peer and financial support.

The decrease in the use of EBP could be explained by several individual and organizational factors. Findings from the current study and those from previous studies [58–62] converge to suggest that practitioners highly value their own personal practice experiences and those of their peers as a primary source of knowledge in making clinical decisions. 

Pages 18-19, lines 343-357

Reviewer 2: -P. 17, line 327. Several of the references 56-60 are not studies of ATs and PTs so should not be used to support a statement that OTs and PTs highly value personal practice experiences….

Authors: We have now omitted OTs and PTs from the line and kept “practitioners” only to make it more generic.

Page 19, line 368

Reviewer 2: -As a continuation of the point above, I do miss a comparison of findings among ATs and PTs with other healthcare professions such as nurses and physicians. Much EBP research has been done also among those professions.

Authors: We agree with the reviewer that there is a substantive and growing body of research available in other healthcare professions; however, our focus was on OTs and PTs only for the reasons provided in the introduction. It is beyond the scope of our study to draw comparisons with other healthcare professions. This would also unnecessarily lengthen the paper. 

Reviewer 2: -P. 16, line 323. Please change wording to ”EBP evolves among” instead of ”…in”.

Authors: We have now made this change.

Page 18, line 344

Reviewer 2: -P. 16, line 327-329. The finding that practitioners value personal practice experience and that of their peers – I can’t see this finding reported in the Results section.

Authors: This finding is in S3 Appendix. 

Reviewer 2: -P. 17, line 339-341. Is this a finding from your study or from previous literature? If the former, I don’t see it reported in the Results section, and if the latter, it needs a reference. 

Authors: This is an interpretation of our qualitative findings which are found in S3 Appendix under three domains: knowledge, social influences and behavioral regulation. 

Reviewer 2: -Please discuss the gender distribution in both the survey data (83%-91%) and the focus group data (87%).

Authors: We intentionally did not discuss gender distribution as we do not consider gender an influencing variable in EBP. Considering the scope and complexity of our study, we intended to focus on discussing those variables that are relevant to EBP. 

Reviewer 2: -Please discuss limitations and strengths of the study.

Authors: Thank you for highlighting this. We added a new section on Strengths and Limitations of this study on Pages 23 and 24. 

Pages 23-24, lines 471-491

Reviewer 2: -Please briefly discuss generalizability and transferability of the findings, as well as reliability, credibility and trustworthiness.

Authors: We have now discussed this in the Strengths and Limitations section. 

Pages 23-24, lines 471-491

Reviewer 2: Conclusion - P. 21, line 440. I believe ”positive” should precede ”attitudes toward EBP”.

Authors: Thank you for pointing it out. We corrected this typo. 

Page 24, lines 494-496

Reviewer 2: References

-Please double check ref 41. I believe there is an ”In” missing to clarify that it’s a book chapter, and the book title is abbreviated so that it seems like a journal title. 

Authors: We corrected this reference. It is now reference # 40. The reference reads as follows:

Creswell JW, Plano Clark VL, Gutmann ML, Hanson WE. Advanced mixed methods research designs. In: Tashakkori SA, Teddlie C, Editors. Handbook of mixed methods in social and behavioral research. Sage Publications. 2003:209-40. 

Page 31, lines 620-622

References:

Birt, L., Scott, S., Cavers, D., Campbell, C., & Walter, F. (2016). Member Checking: A Tool to Enhance Trustworthiness or Merely a Nod to Validation? Qualitative Health Research, 26(13), 1802–1811. https://doi.org/10.1177/1049732316654870

Bussières, A. E., Al Zoubi, F., Quon, J. A., Ahmed, S., Thomas, A., Stuber, K., Sajko, S., & French, S. (2015). Fast tracking the design of theory-based KT interventions through a consensus process. Implementation Science, 10(1), 1–14. https://doi.org/10.1186/s13012-015-0213-5

Bussieres, A., Patey, A., Francis, J., Sales, A., Grimshaw, J., & Team t, C. P. P. (2012). Identifying factors likely to influence compliance with diagnostic imaging guideline recommendations for spine disorders among chiropractors in North America: A focus group study using the Theoretical Domains Framework. Implementation Science, 7(1), 82.

Cane, J., O’Connor, D., & Michie, S. (2012). Validation of the theoretical domains framework for use in behaviour change and implementation research. Implementation Science, 7(1), 37.

Loughran, T., & Nagin, D. S. (2006). Finite sample effects in group-based trajectory models. Sociological Methods and Research, 35(2), 250–278. https://doi.org/10.1177/0049124106292292

McGowan, L. J., Powell, R., & French, D. P. (2020). How can use of the Theoretical Domains Framework be optimized in qualitative research? A rapid systematic review. British Journal of Health Psychology, 25(3), 677–694.

Michie, S., Johnston, M., Lawton, R., Parker, D., & Walker, A. (2005). Making psychological theory useful for implementing evidence based practice: A consensus approach. Qual Saf Health Care, 14, 26–33.

Nagin, D. S., & Nagin, D. (2005). Group-based modeling of development. Harvard University Press.

Roberts, N., Lorencatto, F., Manson, J., & Jansen, J. (2015). Identifying the barriers and facilitators to transforming a university hospital into a Major Trauma Centre: A qualitative case study using the Theoretical Domains Framework. Scandinavian Journal of Trauma, Resuscitation and Emergency Medicine, 23(2), A10. https://doi.org/10.1186/1757-7241-23-S2-A10

Rochette, A., Brousseau, M., Vachon, B., Engels, C., Amari, F., & Thomas, A. (2020). What occupational therapists’ say about their competencies’ enactment, maintenance and development in practice? A two-phase mixed methods study. BMC Medical Education, 20(1), 1–14. https://doi.org/10.1186/s12909-020-02087-4

To, D., Hall, A., Bussières, A., French, S. D., Lawrence, R., Pike, A., Patey, A. M., Brake-Patten, D., O’Keefe, L., Elliott, B., & De Carvalho, D. (2022). Exploring factors influencing chiropractors’ adherence to radiographic guidelines for low back pain using the Theoretical Domains Framework. Chiropractic & Manual Therapies, 30(1), 23. https://doi.org/10.1186/s12998-022-00433-5

---

## [Decision Letter · Decision Letter 1]

20 Mar 2023

Exploring if and how evidence-based practice of occupational and physical therapists evolves over time: A longitudinal mixed methods national study

PONE-D-22-19314R1

Dear Dr. Thomas,

We’re pleased to inform you that your manuscript has been judged scientifically suitable for publication and will be formally accepted for publication once it meets all outstanding technical requirements.

Kind regards,

Yu-Wei Ryan Chen, PhD

Academic Editor

PLOS ONE

Additional Editor Comments (optional):

Reviewers' comments:

Reviewer's Responses to Questions

**Comments to the Author**

1. If the authors have adequately addressed your comments raised in a previous round of review and you feel that this manuscript is now acceptable for publication, you may indicate that here to bypass the “Comments to the Author” section, enter your conflict of interest statement in the “Confidential to Editor” section, and submit your "Accept" recommendation.

Reviewer #1: All comments have been addressed

Reviewer #2: All comments have been addressed

2. Is the manuscript technically sound, and do the data support the conclusions?

Reviewer #1: Yes

Reviewer #2: (No Response)

3. Has the statistical analysis been performed appropriately and rigorously? 

Reviewer #1: Yes

Reviewer #2: (No Response)

4. Have the authors made all data underlying the findings in their manuscript fully available?

Reviewer #1: Yes

Reviewer #2: (No Response)

5. Is the manuscript presented in an intelligible fashion and written in standard English?

Reviewer #1: Yes

Reviewer #2: (No Response)

6. Review Comments to the Author

Reviewer #1: Thank you for the opportunity to review this interesting manuscript. I have limited comments:

Abstract:

No Comments.

Introduction:

The purpose and value of the study well argued. Authors provide convincing arguments within the context of the existing literature for the scope and value of the study.

Objectives

No Comments.

Methods:

Implemented methods are described in detail. The use of the COREQ Guideline is clearly described in the appendix.

l. 172-174: Please consider clarifying this sentence ex.: “Since we wanted to observe the pattern of EBP over time, only those OT and PT graduates who had participated in the baseline survey and were working at the time of data collection were eligible to participate in the subsequent rounds (T1, T2 and T3).”

Results:

S4 Appendix - Please consider the use of identifiers for respondents. Respondent-identifiers are not used under T1 “Knowledge” and T1-3 “Behavioral regulation”, and T2-3 “Skills”. In T1 “Belief about consequences”, T2 “Environmental context and resources”, T2 “Social influences”, T1-2 “Goals” and T1 “Social/Professional role and Identity” respondent identifiers are missing (P?). T1 “Skills”, T2 “Emotions”, T2 “Decision Process”, T1 “Reinforcement”, T1 “Optimism”, and T2 “Intention” use identifiers “PD”, “PB”, “PC”, and “PA”.

Discussion:

Authors clearly discuss study results and support their discussion with the results of their study within the context of the available literature. Authors contextualize study results with recommendations for future research, clinical practice and provide recommendations for education of future OT and PTs.

Reviewer #2: (No Response)

7. PLOS authors have the option to publish the peer review history of their article (what does this mean?). If published, this will include your full peer review and any attached files.

Reviewer #1: No

Reviewer #2: No

---

## [Editor Report · Acceptance letter]

23 Mar 2023

PONE-D-22-19314R1 

Exploring if and how evidence-based practice of occupational and physical therapists evolves over time: A longitudinal mixed methods national study 

Dear Dr. Thomas:

I'm pleased to inform you that your manuscript has been deemed suitable for publication in PLOS ONE. Congratulations! Your manuscript is now with our production department. 

Kind regards, 

on behalf of

Dr. Yu-Wei Ryan Chen 

Academic Editor

PLOS ONE